# Spatiotemporal Variations in Physicochemical and Biological Properties of Surface Water Using Statistical Analyses in Vinh Long Province, Vietnam

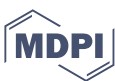

**Nguyen Thanh Giao *** , **Truong Hoang Dan, Duong Van Ni, Phan Kim Anh and Huynh Thi Hong Nhien ***

Department of Environmental Management, College of Environment and Natural Resources, Can Tho University, Can Tho City 900000, Vietnam; thdan@ctu.edu.vn (T.H.D.); dvni@ctu.edu.vn (D.V.N.); pkanh.96@gmail.com (P.K.A.)
* Correspondence: ntgiao@ctu.edu.vn (N.T.G.); hthnhien@ctu.edu.vn (H.T.H.N.)

**Abstract:** In this study, spatiotemporal fluctuations in surface water quality in Vinh Long province, Vietnam, were conducted using entropy weighting, water quality index (WQI), and multivariate statistical techniques, such as cluster analysis (CA), principal component analysis (PCA), and discriminant analysis (DA). The samples collected at 63 monitoring locations in March, June, and September were measured for 15 parameters. Compared to the Vietnamese standard, surface water was contaminated with organic matters, nutrients, microorganisms, and salinity. DA identified the most typical parameters (pH, turbidity, TSS, EC, DO, Cl$^-$, *E. coli*, coliform) in distinguishing temporal variations in water quality with greater than 75% of the correction. CA group 63 sampling sites into 22 clusters representing different land use patterns. WQI determined the worst water quality was found in the agricultural areas. Based on the results of entropy weighting, EC, coliform, N-NH$_4^+$, BOD, N-NO$_3^-$, and Fe had significantly controlled surface water quality. Four principal components obtained from PCA explained 66.45% of the variance, suggesting the influences of geohydrological factors and anthropogenic activities, such as domestic, market area, agriculture, and industry. The findings of this study can provide useful information for authorities to evaluate the effectiveness of monitoring systems and plan for water quality management strategies.

**Keywords:** entropy weight; multivariate statistical techniques; surface water; water quality index

## 1. Introduction

Due to the growth of economic development and living standards, the demand for clean water has continuously increased. Surface water constitutes an important water source serving different human activities because of its abundance and accessibility, especially in the Mekong Delta region. However, despite its integral role in human civilization, its quality has been jeopardized by climate changes and anthropogenic activities [1]. For example, a lack of management in wastewater discharges from domestic, industrial, and aquacultural sources has deteriorated surface water quality [2,3]. In addition, agricultural runoff containing pesticides and herbicides is also a stressor to the quality of surrounding water bodies. According to the Mekong River Commission [4], areas of higher population density and production activities are closely associated with surface water quality degradation. Furthermore, land use has also been confirmed to significantly impact spatial variations in water quality [5]. Consequently, poor water quality and sanitation are linked to serious diseases in humans. It is estimated that over 3.5 million people die annually from waterborne diseases and the majority are children (2.2 million) [6]. Therefore, monitoring surface water quality is of importance to reduce potential risks to environmental problems and public health.

According to the Environmental Protection Law, environmental monitoring programs have been established and implemented in Vietnam [7]. However, the environmental

quality is evaluated by single pass-fail criteria for single parameters in the national standards [8,9]. Moreover, this work is only conducted by environmental management agencies without the participation of scientists. It can be seen that, although huge data related to different components in the environment was generated annually, the current evaluation approaches are very limited and inefficient. Thus, it is required to have studies on the applications of novel approaches to comprehend environmental quality assessment. Multivariate statistical techniques are widely applied to provide representative information of a large amount of information [10–12]. These techniques help identify factors that affect the quality and elaborate spatiotemporal variations in surface water quality. In addition, the water quality index can also be widely applied to assess the suitability of water for different purposes [8]. However, the methods may have certain limitations, so the simultaneous application of multiple assessment methods can provide a comprehensive view of surface water quality dynamics.

Vinh Long province, situated in the Mekong Delta, Vietnam, is an agricultural center specializing in rice and fruit crops and high-tech agriculture. In addition to the agricultural strength, the province is facilitated to develop multi-industry, eco-tourism, and commercial activities. Surface water in Vinh Long, with other provinces in the Mekong Delta, such as Dong Thap and Tien Giang, was polluted, resulting in eutrophication [13]. Urban wastewater in the province was polluted by organic matters, nutrients, and microorganisms [14]. However, surface water assessment with the support of multivariate statistical methods and water quality index has not received much attention, which results in the lack of detailed information for the protection and management strategies. Therefore, the objectives of this study were to determine: (1) the physicochemical and microbial characteristics of surface water by time and space, (2) the specific pollution problems and the correlation between them, and (3) the classification of surface quality and correlation between land use patterns and water quality.

## 2. Materials and Methods

### 2.1. Study Areas

Vinh Long province has an area of 1525.73 km$^2$, with a population of 1,050,241 people. It includes 1 city, 1 town, and 6 districts. The topography is relatively flat, and the elevation is low compared to sea level. With the characteristics of the estuary floodplain, the terrain is in the form of a basin in the center and gradually rises towards the banks of the Tien and Hau rivers and other major rivers. Acid sulfate soils are dominant, even in deeper soil layers, so the pH is low. Fertile soils are mainly concentrated along the Hau and Tien rivers. Because it is located between the two largest rivers in the Mekong Delta (Tien and Hau rivers), the network of rivers are interlaced, forming a fairly complete natural water distribution system. In addition to this system, the large average annual rainfall has created favorable conditions for domestic and production activities. Water resources in the province mainly rely on the continental surface water of 91 river and canal systems. Surface water sources are relatively evenly distributed. Water influents for water treatment plants in the area are from major rivers, such as Tien, Hau, Co Chien, and Mang Thit Rivers. Agricultural production and agricultural economy are considered strengths of the province. Additionally, the province has 2 industrial zones and 3 other planning zones. In the context of economic development, surface water quality in this area has been seriously threatened.

### 2.2. Water Sample Collection and Analysis

Sixty-three surface water samples were collected in three periods (March, June, and September) from different locations in Vinh Long province, Vietnam, as depicted in Figure 1. These locations investigated surface water in urban areas, densely populated areas, central market areas, industrial parks, and handicraft villages with more than 30 rivers/channels. There are 4 main rivers: Tien River (labeled as S1, S30, S49, S54), Co Chien River (S4, S10, S41, S47, S48), Mang Thit River (S5, S20, S21, S22, S37, S61), and Hau River (S23, S24, S25, S39, S40, S42, S55). Level 2 and 3 river/canal systems include: Cai Da River (S2, S3); Cai

Doi River (S6, S7); Cai Cam River (S8, S9); Cau Vong River (S11); Long Ho River (S12, S13, S14, S16); An Hiep River (S17); Ong Me Nho River (S18); Ba Lang River (S19 and S44); Cai Von River (S26); Ba Cang river (S27); Bung Truong river (S28); Vung Liem River (S29 and S32); Tra Ngoa River (S31); Thay Cai Channel (S33); Hoa My River (S34, S35); Cai Ngang River (S36); Loc Hoa River (S43); Bo Ke river-Ba Lang river (S45 and S46); Loc Hoa river and canal No. 4 (S50); Cai Con river (S51 and S52); Nha Man River-Tu Load (S53); Tra Son River (S56); Xa Khanh river (S57); Nga Chanh River (S58); May Phop River (S59); Ngai Tu river (S60); Xang channel (S62); Loan My River (S63).

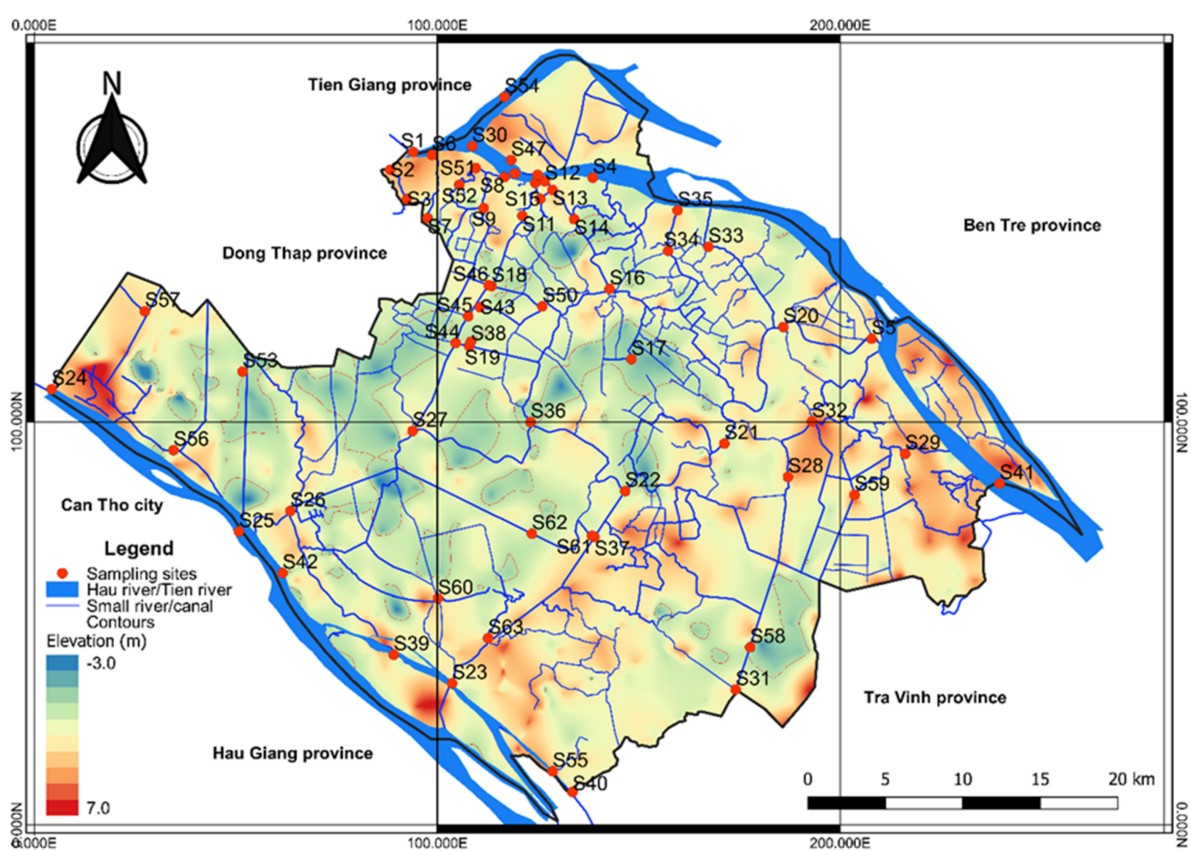

**Figure 1.** Locations of surface water monitoring in Vinh Long province.

A pooled water sample collected at each monitoring location depth 30–50 cm below the surface was preserved in clean plastic bottles. Fifteen parameters were used to evaluate water quality: pH, temperature (°C), turbidity, total suspended solids (TSS), electrical conductivity (EC), dissolved oxygen (DO), biochemical oxygen demand (BOD), chemical oxygen demand (COD), ammonium (N-NH$_4^+$), nitrate (N-NO$_3^-$), orthophosphate (P-PO$_4^{3-}$), chloride (Cl$^-$), *E.coli*, coliforms, and iron (Fe). The physicochemical parameters, including pH, temperature, DO, turbidity, and EC were determined on-site by using a portable pH meter (pH HQ 11D-Hach), DO meter (HANNA HI9146-Hanna, Romania– Hanna, Romania), turbidity meter (HANNA HI93703-Hanna, Romania), and conductivity meter (HANNA HI98318). In the laboratory, 10 parameters (TSS, BOD, COD, N-NH$_4^+$, N-NO$_3^-$, P-PO$_4^{3-}$, Cl$^-$, Fe, *E. coli*, and coliform) were analyzed according to standard methods [15]. All the water parameters were analyzed by the laboratories accredited according to ISO/IEC 17025:2017 with the code number Vilas 515.

*2.3. Data Processing*

The collected water quality data was processed by Microsoft Excel version 2016 software to standardize data and descriptive statistics (minimum, maximum, mean, and standard deviation). After that, the results were presented as box-plot charts.

2.3.1. Integrated Water Quality Assessment Method

The overall water quality assessment method was implemented based on relatively fixed water monitoring variables. In this study, the water quality assessment index was calculated according to Decision No. 1460/QD-TCMT on the issuance of technical guidelines for the calculation of water quality index in Vietnam (WQI) [16]. The parameters used to calculate WQI include pH (Group I-WQI$_I$); DO, BOD, COD, N-NH$_4^+$, N-NO$_3^-$, P-PO$_4^{3-}$ (Group II-WQI$_{II}$); *E. coli*, and coliform (WQI$_{III}$). The final WQI calculation in this study was performed using Equation (1).

$$\text{VN}_{\text{WQI}} = \frac{\text{WQI}_I}{100} \times \left(\frac{1}{6}\sum \text{WQI}_{II} \times \frac{1}{2}\sum \text{WQI}_{III}\right)^{\frac{1}{2}} \tag{1}$$

Water quality is divided into 6 levels based on the values of WQI: very good (WQI = 91–100), good (WQI = 76–90), moderate (WQI = 51–75), bad (WQI = 26–50), poor (WQI = 10–25), and heavily polluted (WQI < 10). The WQI values of each location were presented on the land cover classification map so that the overall water quality could be visually assessed.

In addition, the relative correlation between land use patterns and water quality was determined. This land cover classification was performed by interpreting Landsat 8 images using QGIS version 3.16 software (the Open Source Geospatial Foundation-OSGeo, Chicago, IL, USA). Each land cover type was assigned an identification with a certain color for proper distinction: (1) Water surface, (2) Residential area, (3) Agricultural land, and (4) Non-agricultural land.

2.3.2. Calculation of Entropy Weight

Each water quality parameter has its distinct contribution to water quality. Therefore, the weight of each water quality parameter is calculated to determine its importance, which can be consequently used in ranking the effect of these parameters on surface water. In the study, the weights of the parameters were calculated using the entropy weight method [17–19]. The u eigenvalue matrix data is associated with m water samples and *n* parameters in Equation (2).

$$u_{ij} = \begin{bmatrix} u_{11} & u_{12} & u_{1m} \\ u_{21} & u_{22} & u_{2m} \\ u_{31} & u_{32} & u_{3\,m} \\ \ldots & \ldots & \ldots \\ u_{n1} & u_{n2} & u_{nm} \end{bmatrix} \tag{2}$$

where i is the water quality parameter (i = 1, 2, 3, ... , m); j is the study sites (j = 1, 2, 3, ... , *n*); m and *n* are the number of parameters and study sites, respectively. The calculation of Cl$^-$ was excluded since this indicator is locally affected in a very large set of data, which could affect the calculation for the entire study area.

For a given water quality dataset, the data normalization function of the u$_{ij}$ matrix was calculated using Equation (3). This step is to eliminate the influence of different units and sample numbers of water quality parameters.

$$v_{ij} = (u_{ij} - \min(u_{ij}))/(\max(u_{ij}) - \min(u_{ij})) \tag{3}$$

Then, the information entropy (E$_i$) of the parameter i is determined by Equations (4)–(6). The smaller the value of E$_i$, the greater the influence of parameter i.

$$E_i = -(1/\ln n) \times \sum_{j=1}^{n} S_{ij}.\ln S_{ij} \tag{4}$$

$$S_{ij} = v_{ij}/\sum v_{ij} \tag{5}$$

If $S_{ij} = 0$ or $S_{ij} = 1$; $S_{ij}$ is calculated by the Equation (6).

$$S_{ij} = (1 + v_{ij}) / \sum_{j=1}^{n} (1 + v_{ij}) \tag{6}$$

Finally, the weight of parameter i can be calculated by Equation (7).

$$w_i = (1 - E_i) / (m - \sum_{i=1}^{m} E_i), \ w_i \in [0, 1] \ \text{và} \ \sum_{i=1}^{m} w_i = 1 \tag{7}$$

To determine the relationship of water quality parameters with the highest pollution contribution, Pearson correlation analysis was performed on Statgraphics Centurion version XVI (Statgraphics Technologies Inc., The Plains, VA, USA). Pearson correlation analysis was considered significant when $p < 0.05$ and higher when $p < 0.01$. The quantification of these relationships is expressed through the correlation coefficient (r). Assuming X and Y are the content of two water quality parameters, the r value is calculated using Equation (8) [11]. The larger the r value, the more correlated the parameter is, with the value $r \in [-1, \ 1]$.

$$r = \sum (X - \overline{X}).(Y - \overline{Y}) / \sqrt{\sum (X - \overline{X})^2.(Y - \overline{Y})^2} \tag{8}$$

2.3.3. Multivariate Statistical Methods

Discriminant analysis (DA) was used to evaluate the temporal variation of water quality parameters. In this study, DA was applied to a raw data matrix using both standard and stepwise modes to establish discriminant functions (DFs) to distinguish variations in water quality. Temporal-based DA analysis was performed after dividing the entire data set into three groups (March, June, and September). The results of the DA analysis were able to identify key parameters that led to statistically significant seasonal differences at all sites. Furthermore, cluster analysis (CA) was applied in grouping study sites to classify water bodies in the study area using Euclidean distance. The result of CA was reported as link distance, and the distance between clusters is considered to have clustering significance when $D_{link}/D_{max} \times 100 = 60$ [20]. Sampling locations were considered group variables (dependent variables), and parameters were considered independent variables.

Moreover, the principal component analysis (PCA) has also been applied to determine the principal components (PCs) responsible for variations in water quality in the study area. Each PC would be considered a potential source of pollution by looking at the correlation between water quality parameters [20,21]. The Eigenvalue coefficient can measure the importance of PC, which corresponds to the importance of the source of water quality fluctuations. The larger the coefficient, the more the source of water quality variation contributes to the water quality variation, and the most important components are determined with the Eigenvalues of greater than 1 [20,22]. In addition, the weighted correlation coefficient of each PC can support identifying pollution sources and is rated at three levels of high, moderate, and weak, with absolute load values >0.75, 0.75–0.50, and 0.50–0.30, respectively [23]. These analyzes were performed using Statgraphics Centurion version XVI software (Statgraphics Technologies Inc., The Plains, VA, USA).

**3. Results**

*3.1. Water Quality Characteristics in Vinh Long Province*

The results of physicochemical and microbial properties of surface water in Vinh Long province are depicted in Figure 2. The temperature was measured from 28.3–31.43 °C, with an annual average of 30.13 ± 0.95 °C. pH in surface water ranged from 7.08–7.94, with an average of 7.50 ± 0.3, which was within the Vietnamese standard, column A1 (QCVN 08-MT:2015/BTNMT) [24]. The findings of EC varied from 16.00–51.40 mS/m, with the lowest and highest values at S8 and S58. The average EC was about 27.04 ± 16.50 mS/m. TSS concentration at all monitoring locations exceeded the standard (20 mg/L) from 1.13 to 2.48 times, with an average of 35.51 ± 8.92 mg/L. The highest TSS (49.67 mg/L) was found

at S13, and the lowest (22.67 mg/L) was at S62. On the other hand, TSS concentration also had a certain positive correlation with the turbidity value, with values ranging from 26.10–95.70 NTU, with an average of 52.61 ± 28.53 NTU.

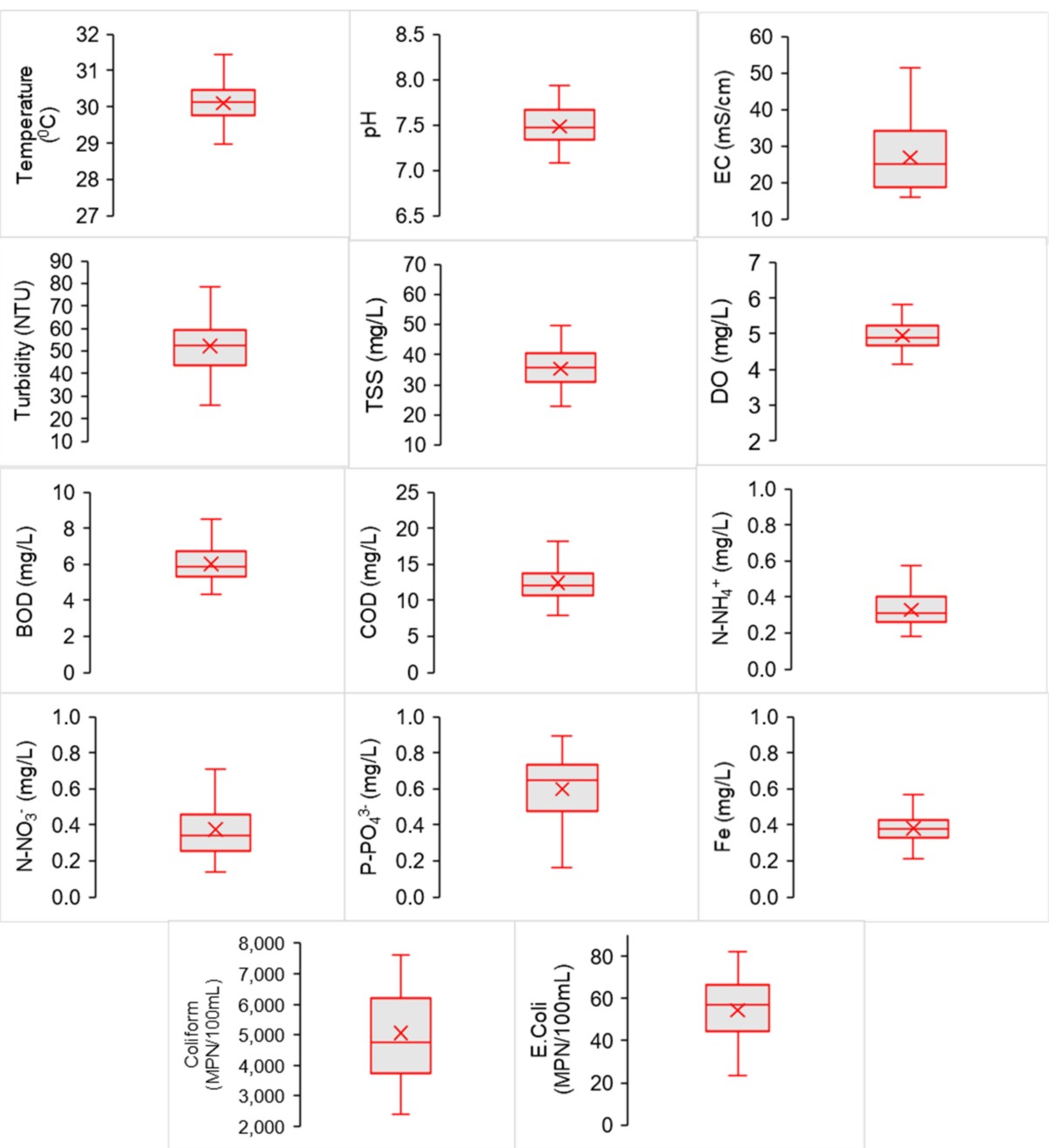

**Figure 2.** Boxplot of water quality parameters in Vinh Long province.

DO concentration fluctuated from 4.14–6.10 mg/L, with an average of 4.98 ± 0.65 mg/L. The findings of DO concentration found at most of the monitoring sites were below the standard ($\geq$6 mg/L). Only 2 out of 63 positions (S1 and S3) met the standard, accounting for 3.17% of the total. The low DO in the study area was consistent with the high BOD and COD concentrations. The values of BOD and COD ranged from 4.35–8.50 and 7.92–18.20 mg/L, respectively, which also exceeded the standard (4 mg/L-BOD and 10 mg/L-COD). BOD pollution was recorded at all monitoring locations, whereas 87% of the total with COD pollution was observed.

The concentrations of N-NH$_4^+$, N-NO$_3^-$, and P-PO$_4^{3-}$ fluctuated in the range 0.18–0.58, 0.14–0.71, and 0.16–0.89 mg/L, respectively. As presented in Figure 2, P-PO$_4^{3-}$ concentration was over the standard (0.1 mg/L) at all locations. While most locations were found with N-NH$_4^+$ pollution in surface water (>0.3 mg/L), N-NO$_3^-$ was still within the standard (2 mg/L).

*E.coli* density varied considerably from 23.67–82.00 MPN/100 mL, with an average of 54.61 ± 16.80 MPN/100 mL. Similarly, high coliform density was detected with a range of 2400–17,466.67 MPN/100 mL, and its average was 5073.02 ± 3279.62 MPN/100 mL. The densities of *E.coli* and coliform were 1.18–4 times higher than the allowable limits (20 MPN/100 mL for *E. coli* and 2500 MPN/100 mL for coliform). The highest position recorded was nearly seven times higher than the limit.

Cl$^-$ concentration was not detected at most sites, except for S28, S31, S40, and S41, and only S41 was recorded to have exceeded the allowable limit (200 mg/L). Fe concentration was also detected at most of the sites; however, only about 10% of sampling sites exceeded the allowable limit (0.5 mg/L). Fe concentration at 63 monitoring locations ranged from 0.21–0.67 mg/L, with the lowest and highest at S04 and S51. An average of Fe concentration was about 0.38 ± 0.99 mg/L.

### 3.2. Temporal Variations in Surface Water Quality

The changes in surface water quality by season in the study area were evaluated using DA that divided the dataset into three time intervals (March, June, and September) (Table 1). The results of DF were constructed by standard modes, forward stepwise and backward stepwise. They obtained classification matrices by 79.37%, 75.13%, and 77.25% of cases, contributing 15, 9, and 8 parameters, respectively (Table 2).

**Table 1.** Coefficients of functions discriminant to the temporal variations.

| Methods | Function | Eigenvalue | %Variation | Canonical Correlation | Wilks Lambda | Chi-Square | *p*-Value |
|---|---|---|---|---|---|---|---|
| Standard | 1 | 1.73 | 90.64 | 0.80 | 0.31 | 209.32 | 0.00 |
| | 2 | 0.18 | 9.36 | 0.39 | 0.85 | 29.45 | 0.01 |
| Forward stepwise | 1 | 1.69 | 91.50 | 0.79 | 0.32 | 206.82 | 0.00 |
| | 2 | 0.16 | 8.50 | 0.37 | 0.86 | 26.57 | 0.00 |
| Backward stepwise | 1 | 1.71 | 91.17 | 0.79 | 0.32 | 208.62 | 0.00 |
| | 2 | 0.17 | 8.81 | 0.38 | 0.86 | 27.80 | 0.00 |

The standard mode recorded that Eigenvalues were 1.73 and 0.18, which accounted for 90.64 and 9.36% of the variance that explained the cause of the surface variation (Table 1). The canonical correlation coefficients were 0.80 and 0.39, respectively, indicating that 20.4% and 61.1% of the 15 parameters were explained by this analysis. The forward stepwise determined nine parameters (pH, turbidity, TSS, EC, COD, DO, Cl$^-$, *E. coli*, coliform) that control temporal variations in surface water quality. Meanwhile, only eight parameters obtained from the backward stepwise method cause these variations. The seasonal trends of the nine identified parameters are shown in Figure 3. Specifically, pH, turbidity, and TSS in September (rainy season) were higher than those in March (dry season) and June (early rainy season) (Figure 3). In contrast, the highest concentrations of DO, *E.coli*, and coliform were determined in March, while only EC was highest in June.

**Table 2.** Classification functions for discriminant analysis of temporal variations.

| Parameter | Standard | | | Forward Stepwise | | | Backward Stepwise | | |
|---|---|---|---|---|---|---|---|---|---|
| | **March** | **June** | **September** | **March** | **June** | **September** | **March** | **June** | **September** |
| pH | 68.08 | 66.83 | 68.63 | 62.14 | 60.85 | 62.78 | 62.53 | 61.26 | 63.16 |
| Temp | 36.42 | 36.57 | 36.62 | - | - | - | - | - | - |
| Tur | 0.31 | 0.27 | 0.38 | 0.03 | −0.02 | 0.10 | 0.08 | 0.04 | 0.14 |
| TSS | 0.19 | 0.14 | 0.31 | 0.44 | 0.39 | 0.56 | 0.43 | 0.38 | 0.55 |
| EC | 0.73 | 0.80 | 0.69 | 0.34 | 0.40 | 0.29 | 0.37 | 0.43 | 0.32 |
| DO | 15.18 | 14.83 | 13.44 | 12.59 | 12.21 | 10.91 | 12.10 | 11.68 | 10.43 |
| BOD | −1.60 | −1.62 | −1.06 | - | - | - | - | - | - |
| COD | 2.17 | 2.23 | 1.88 | 1.80 | 1.94 | 1.77 | - | - | - |
| $N-NH_4^+$ | −39.68 | −37.57 | −37.92 | - | - | - | - | - | - |
| $N-NO_3$ | 39.95 | 40.13 | 38.68 | - | - | - | - | - | - |
| $P-PO_4^{3-}$ | 19.40 | 21.57 | 20.25 | - | - | - | - | - | - |
| Chloride | 0.06 | 0.07 | 0.05 | 0.09 | 0.09 | 0.08 | 0.06 | 0.07 | 0.06 |
| *E. coli* | −0.09 | −0.14 | −0.13 | 0.25 | 0.21 | 0.22 | 0.34 | 0.31 | 0.32 |
| Coliform | 0.00 | 0.00 | 0.00 | 0.00 | 0.00 | 0.00 | 0.00 | 0.00 | 0.00 |
| Fe | −6.80 | −10.35 | −5.52 | - | - | - | - | - | - |
| Constant | −877.77 | −867.99 | −884.31 | −300.99 | −287.28 | −301.95 | −294.23 | −279.37 | −295.41 |

Notes: Temp: Temperature; Tur: Turbidity.

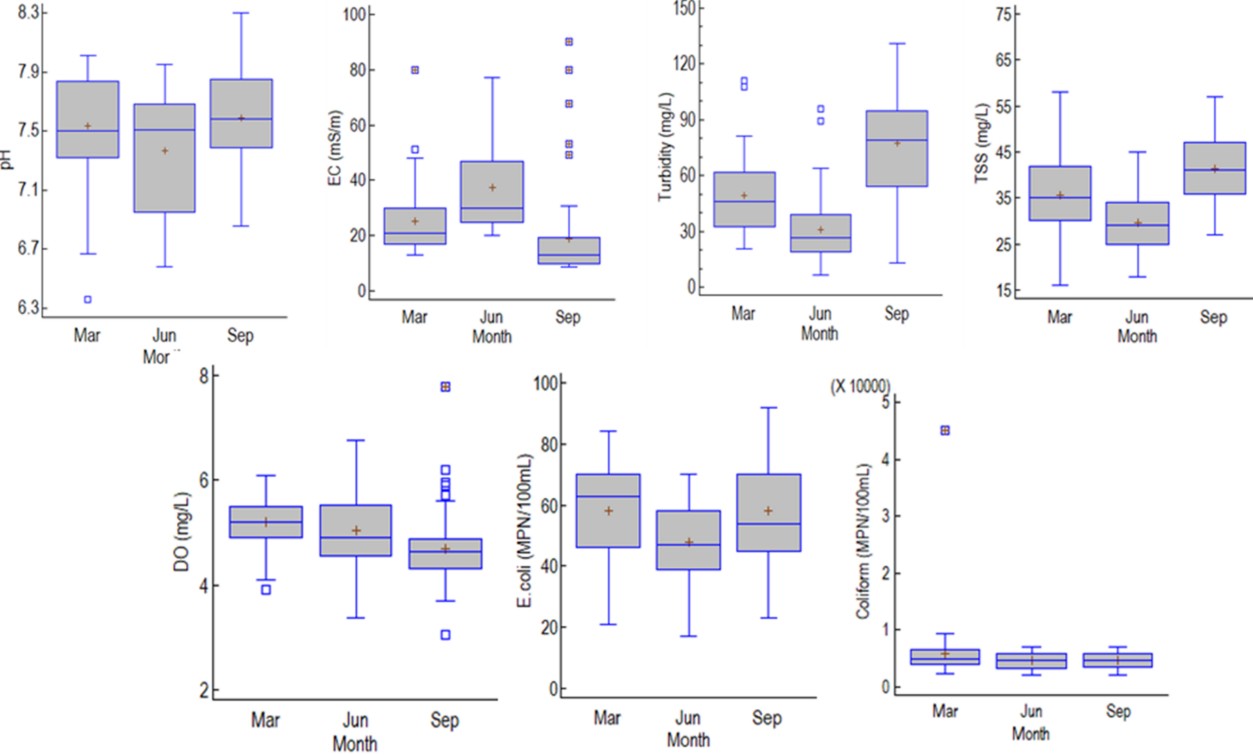

**Figure 3.** Temporal varying water quality parameters.

### 3.3. Clustering Surface Water Quality and Water Quality Index

Figure 4 showed the results of CA with 22 groups established by clustering at a distance $(D_{link}/D_{max}) \times 100 = 60$. Groups 13, 14, 17, and 22 were formed by every distinct location. Group 1 was formed by the positions S1–S3, S6, and S7, while positions S8, S9, S13, and S14 were classified in Group 2. These two groups belonged to the major rivers and tributaries of the Northern part of Vinh Long province. In addition, two secondary tributaries (Long Ho and My Hoa rivers) belonging to the tributary area of Co Chien river were recorded with similar water quality and classified in the same Group 3 (S12, S34, and S35). Notably, the sampling points that were collected on the same rivers/canals or

different rivers/canals but with small areas or located far from the main rivers shared the similar trends in water quality. Specifically, Group 4 was formed from two locations at the beginning and the end of Cai Con river (S51 and S52); the location on Cai Von River (S26) and Tra Son River (S56) (Group 7); Vung Liem river (S29, S32), Ba Lang river (S44), and Ba Cang river (S27) (Group 8); An Hiep River (S17) and Ong Me Nho River (S18) (Group 10); Bung Truong river (S28) and Tra Ngoa river (S31) (Group 12). In addition, samples belonging to inland canals were determined: Group 19 (S36, S43, S46, S50, and S63), Group 20 (S53, S57, S58, and S59), Group 21 (S37, S38, S60, and S61), and Group 22 (S62). Groups 19–22 can be considered nearly identical in water quality because of their short Euclidean distances. Similarly, the positions with the same characteristics were classified into the same group. In addition, the current results can also be used in minimizing the locations with insignificant changes in the water monitoring information required. Specifically, 12 locations on the same river and belonging to the same group (Groups 1–6, 11, 16, and 21) can be removed from the monitoring network.

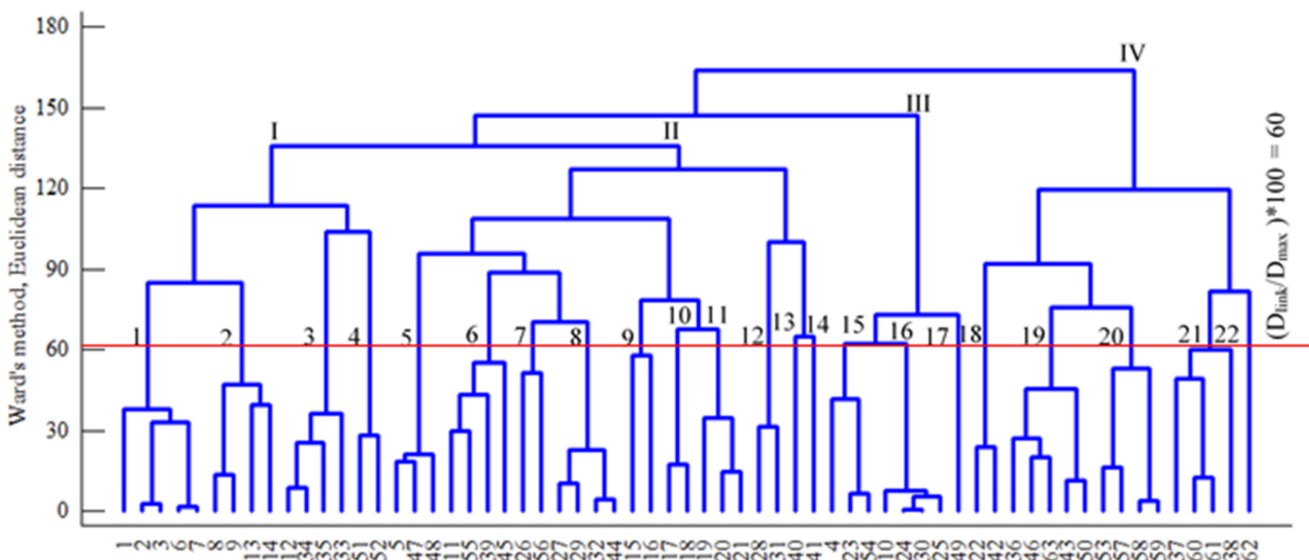

**Figure 4.** Clustering surface water quality in the water bodies in Vinh Long province.

Of the total sampling sites, 38.1% (24 out of 63 sites) had good water quality (76 < WQI < 90) and the majority (57.1%) had medium quality (51 < WQI < 75). Very good water quality was recorded for only about 4.8% (3 out of 63 sites). The spatial variations of surface water quality based on the results of WQI are illustrated in Figure 5. Areas of very good water quality were found at major river locations (the Hau and Co Chien rivers). In addition, areas with poorer water quality were identified, concentrated mainly in locations near main rivers and land with non-agricultural purposes. The medium water quality of the water samples was located in the in-field area, where there were a lot of agricultural activities.

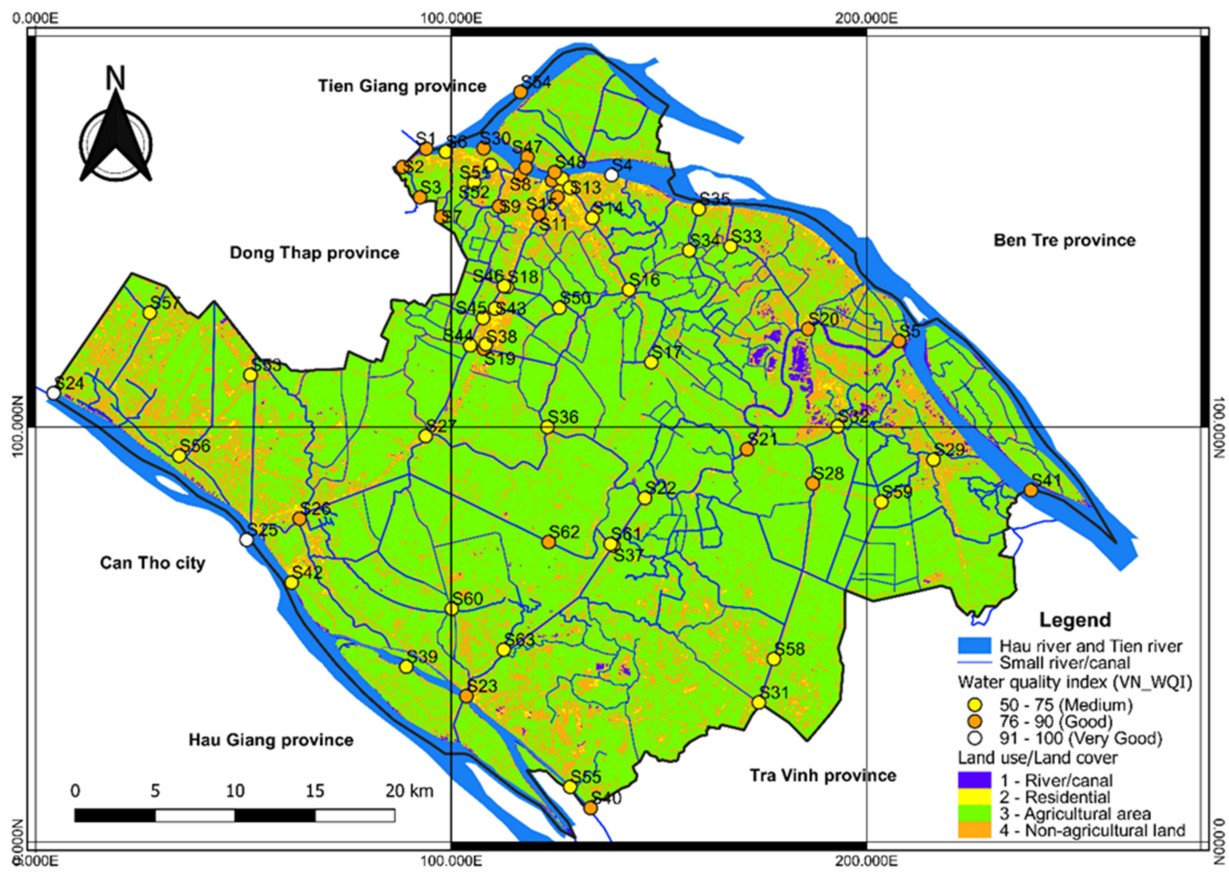

**Figure 5.** Spatial distribution of surface water quality by WQI.

*3.4. Ranking and Correlating Surface Water Quality Parameters*

The results of the entropy weights for the parameters showed that EC, coliform, and N-NH$_4^+$ had the values $\geq 0.1$ (Table 3). This indicated that these parameters primarily contributed to the degradation of surface water quality. The next three parameters with average contributions to water pollution in the study area were BOD, N-NO$_3^-$, and Fe. The temperature had no significant effect. Little impact on water quality deterioration was determined in the remaining parameters. Pearson correlation matrix was used to determine the relationship between six water quality parameters with high entropy values, as illustrated in Figure 6. These parameters showed strong positive correlations with each other. EC showed weak positive correlations with BOD, N-NH$_4^+$, and N-NO$_3^-$. In addition, BOD had positively high correlations with N-NH$_4^+$ (r = 0.75) and coliform (r = 0.60) and weak correlation with N-NO$_3^-$ (r = 0.47). Furthermore, N-NH$_4^+$ was found to be correlated with coliform (r = 0.55) and N-NO$_3^-$ (r = 0.25), whereas N-NO$_3^-$ also showed a moderate positive correlation with coliform (r = 0.56) and Fe (r = 0.31). There was a correlation between coliform and Fe (r = 0.32). However, there was no significant correlation between Fe and EC, BOD, and N-NH$_4^+$.

**Table 3.** Ranking of water pollutants based on the weight of entropy.

| Par. | Temp | pH | Tur | TSS | EC | DO | BOD | COD | N-NH$_4^+$ | N-NO$_3^-$ | P-PO$_4^{3-}$ | *E.coli* | Coliform | Fe |
|------|------|----|----|-----|----|----|-----|-----|-----------|-----------|--------------|---------|----------|----|
| E$_i$ | 1.00 | 0.97 | 0.98 | 0.97 | 0.92 | 0.97 | 0.95 | 0.97 | 0.95 | 0.95 | 0.98 | 0.98 | 0.94 | 0.96 |
| W$_i$ | 0.00 | 0.06 | 0.05 | 0.06 | 0.15 | 0.06 | 0.09 | 0.05 | 0.10 | 0.09 | 0.04 | 0.04 | 0.12 | 0.08 |
| Rank | 9 | 6 | 7 | 6 | 1 | 6 | 4 | 7 | 3 | 4 | 8 | 8 | 2 | 5 |

Notes: Par.: Parameter; Tur: Turbidity.

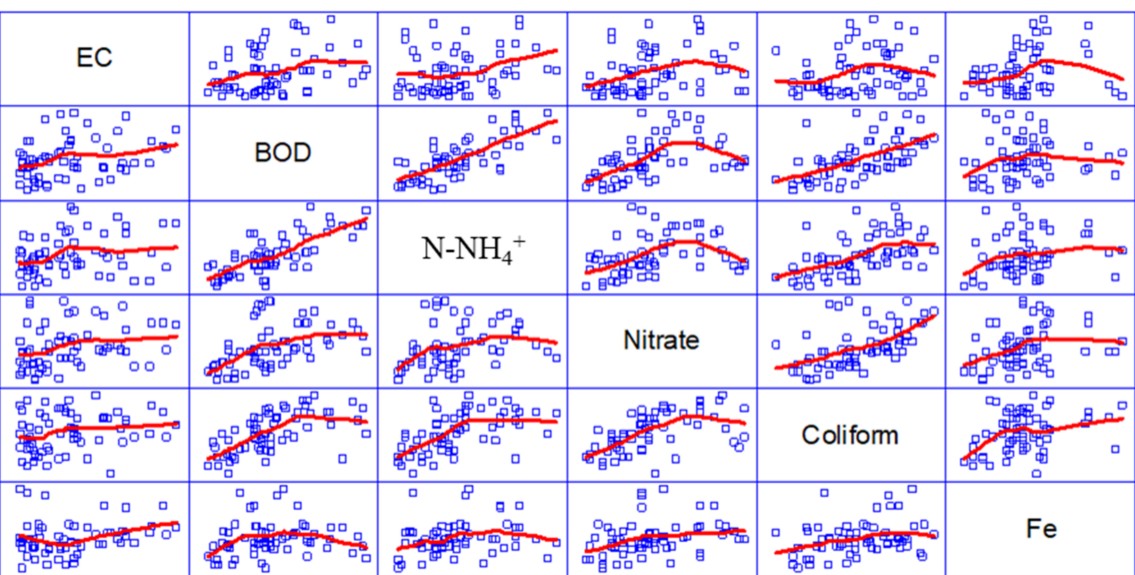

**Figure 6.** Correlation among the most water pollutants.

### 3.5. Key Variables Influencing Surface Water Quality

Four PCs with Eigenvalues greater than 1 obtained from the PCA significantly influenced the variations in surface water quality, as presented in Figures 7 and 8. That can explain 66.45% of the total variance. However, the remaining PCs still have indicators showing the moderate correlation. Thus, it can be seen that the fluctuation of surface water quality in the study area is very complicated and can be affected by several pollution sources. The coordinates of the parameters showed that BOD, COD, $N-NH_4^+$, $N-NO_3^-$, $P-PO_4^{3-}$, *E.coli*, and coliform explained 34.20% for the variation of PC1. PC2 axis indicated that the load factors were positive for TSS, $P-PO_4^{3-}$, and Fe and negative for turbidity, which explained 12.70% of the total variation in water quality. In addition, PC3 showed moderate correlation with pH and $Cl^-$ and weak correlation for DO, BOD, and COD.A total of 11.18% of the total variation in water quality was explained by this PC. Finally, PC4 contributed 8.38% of the variation in water quality with weak correlations with pH, temperature, turbidity, EC, $Cl^-$, and Fe.

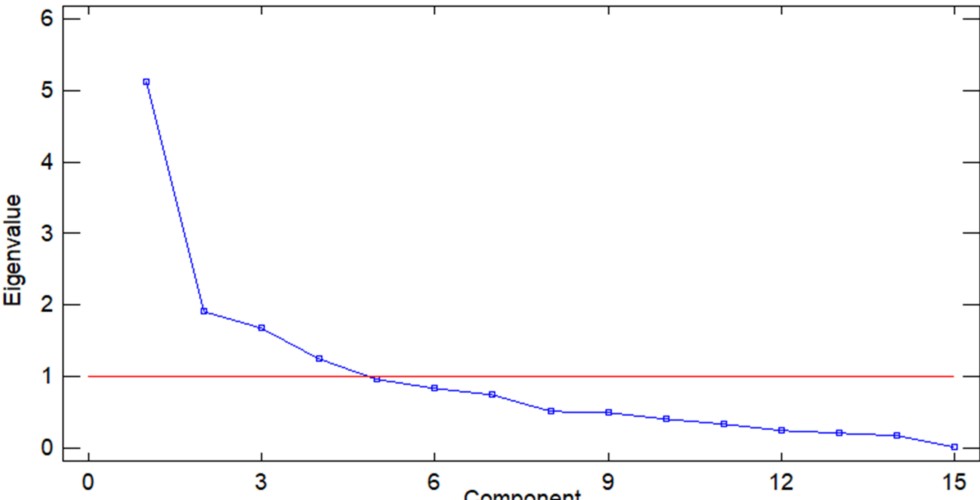

**Figure 7.** Eigenvalues of the PCs.

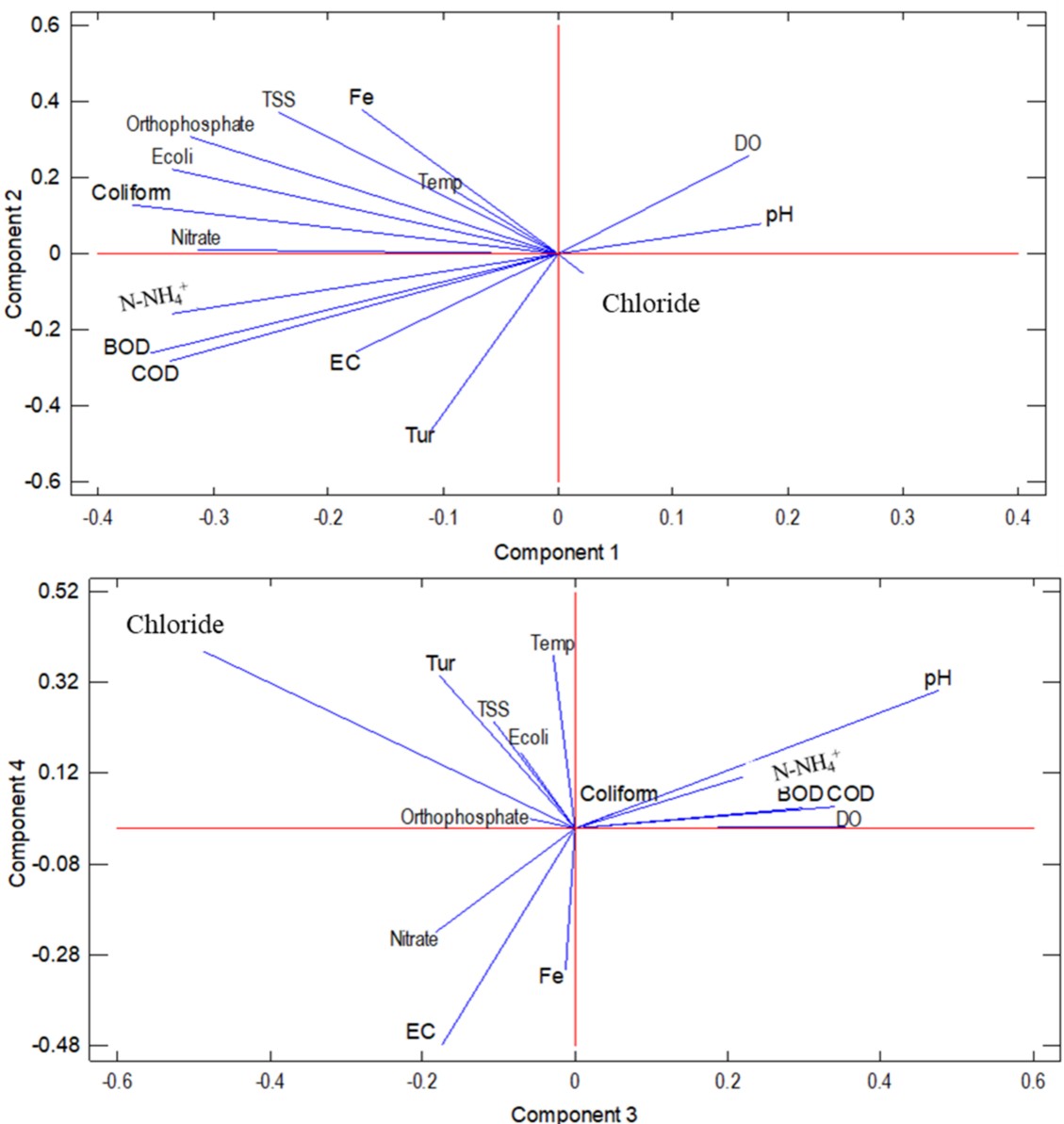

**Figure 8.** Key variables influencing surface water quality in the study area.

## 4. Discussion

### 4.1. Water Quality Characteristics in Vinh Long Province

The results showed that the water pH was generally neutral, but some locations were slightly alkaline. This could increase the toxicity of ammonia in water, thus affecting the life of organisms [25]. pH value in the study is close to that in neighboring provinces, such as Dong Thap (7.00–7.52), Can Tho (7.17–7.53), Hau Giang (6.8–7.1), and Tien Giang (7.2–7.8) [12,26–28]. The findings presented that pH and water temperature in the surface water of Vinh Long are consistent with the evolution of the tropics [23]. EC in Tien Giang province was in the range of 40.59–99.28 mS/m, higher than in the current study area [28], which was associated with the influence of geographic location and flow rate. The suitable EC for fish and large invertebrates in freshwater environments is in the range of 15–50 µS/cm [29]. In general, high EC values in surface water are mostly found in residential areas, industrial zones, and central markets, where a large amount of wastewater contains organic matters, leading to increased mineral segregation.

TSS concentration and turbidity were recorded as quite high in the study area. The turbidity had a high standard deviation due to the high dispersion, resulting in spatial

variation. The turbidity in the water in the rivers and canals of the Mekong Delta varied from 3.25–59.17 NTU [30]. Additionally, TSS concentrations in neighboring provinces ranged from 32.8–101.8 mg/L (Hau Giang), 34.60 mg/L (Dong Thap), and 7.7–121.8 mg/L (Tien Giang) [12,27,28]. It can be seen that the area had lower TSS than neighboring areas. However, higher TSS could increase the cost of water treatment and directly affect aquatic life. TSS is also a carrier that helps transport pollutants, such as microorganisms, pesticides, and antibiotics, to different places, increasing the possibility of human exposure to pollutants.

The DO concentration in the study area has not been found to affect the growth of aquatic organisms significantly. Furthermore, DO was also reported to have a higher value than Hau Giang (3.2–5.2 mg/L), Dong Thap (4.73–5.55 mg/L), and Tien Giang (3.2–4.0 mg/L) L) [12,27,28]. The water surface areas of the river system in Vinh Long are larger than other neighboring provinces, which facilitates the diffusion of oxygen from the air into surface water. In contrast, BOD and COD values indicated that the water bodies were contaminated with organic matter. These problems in surface water in other provinces have been recognized [12,27,28]. Moreover, high BOD and COD concentrations could inversely affect the oxygen concentration in water bodies.

The results showed that the values of nutrients in the study area were at a high level. $NH_4^+$ and $P-PO_4^{3-}$ concentrations were found in Hau Giang province (0.27 and 0.23 mg/L), Dong Thap (0.37 and 0.23 mg/L), and Tien Giang (0.4 mg/L and 0.1 mg/L), respectively [12,27,28]. The dominance of agricultural activities in the regional economy has resulted in high nutrient content. According to Ustaoglu and Tepe [31], $P-PO_4^{3-}$ in surface water ranges from 0.05–0.3 mg/L, and the excess level will cause eutrophication. Thus, the surface water quality in Vinh Long showed signs of nutrient pollution but was not serious. However, in the long term, if there are no solutions to reduce the concentration of $P-PO_4^{3-}$, this will become a serious problem for the aquatic ecosystem.

Coliform and *E.coli* are indications of contamination from human or animal fecal wastes in water bodies. In addition, the relatively high presence of *E. coli* and coliform has resulted in the surface water source in Vinh Long province becoming undrinkable. It is consistent with previous studies in Dong Thap [27], Tien Giang [28], Hau Giang [12], and Can Tho [26].

The concentration of $Cl^-$ was detected only at 4 monitoring locations, in which two locations were detected near the market area, which receives wastewater from not only domestic activities but also productions. The remaining two locations were the confluence points between Vinh Long-Tra Vinh-Soc Trang-Ben Tre provinces, located near the river mouth, thus being strongly affected by the saline intrusion. $Cl^-$ concentrations were also found in other provinces: Dong Thap (9.08–21.84 mg/L) and Tien Giang (6.0–366.4 mg/L) [27,28]. In general, the surface water quality of Vinh Long province did not have saline intrusion, except for S41 area showing signs of saline intrusion. In addition, the locations with relatively high Fe concentrations were also concentrated in the market areas and in the field of agricultural cultivation through the acid washing process, making the water contaminated with Fe.

### 4.2. Temporal Variation of Surface Water Quality

In the case of standard DA, there were a total of 15 parameters to distinguish 79.37% of samples from three different seasons. This designation was reduced by only 4% when nnie parameters were used and 2% when eight parameters were used. Therefore, in this study, DA analysis produced the most satisfactory results for differentiating water quality between the three-time sampling, providing the important parameters without losing the most important parameter by the forward stepwise. Therefore, it can be concluded that pH, turbidity, TSS, EC, DO, $Cl^-$, *E. coli*, and coliform were the most important parameters in distinguishing the monitoring periods in the study area. This result is consistent with the water bodies An Giang province [32]. Therefore, eight parameters showed that the most significant variations in water quality, and they can be prioritized in selecting water quality

monitoring parameters in the future. The feasibility of DA analysis for this backward stepwise was also documented in many previous studies [33,34]. The results of seasonal variations in parameters in the study area can be explained by rainfall and surface runoff. Typically, the influence of water disturbances observed by rainwater and other runoffs is responsible for the significant difference in turbidity and TSS values under different time conditions. This is consistent with the previous finding of Giao et al. [27] that TSS concentration was higher in the rainy months. On the other hand, this surface runoff is also the main route that transfers pollutants from terrestrial to aquatic environments. Additionally, the density of *E.coli* and coliform is high in the dry season. Meanwhile, the high EC values at the beginning of the rainy season can be explained by the leaching of pollutants with high dissolved ion content.

### 4.3. Clustering Surface Water Quality and Water Quality Index

The division of groups from 63 sites showed complex spatial effects of water quality in the study area. Groups 1–4 included the residential areas and large rivers, which were considered the least polluted because of the large flows of rivers. Meanwhile, Groups 6–14 were affected mostly by market areas. However, in Group 13 (S40) and Group 14 (S41), there was a difference in water quality compared to other locations. This can be explained by these two locations located on the Hau and Co Chien rivers, the boundary area between Vinh Long-Tra Vinh-Soc Trang and Vinh Long-Tra Vinh-Ben Tre. On the other hand, Groups 20, 21, and 22 included the agricultural cultivation areas, with small water surface areas and poor flow, where there is the most polluted water quality. The water quality was recorded with almost no significant difference at the locations at the beginning and the end of the rivers. In addition, reducing the number of monitoring locations contributes to simplifying the data set and saves analysis costs and monitoring time but still ensures representativeness of the monitoring task. Effective applications of CA techniques in water quality monitoring design have been reported in other water bodies in the Vietnamese Mekong Delta [12].

The results of the WQI determined that water quality in most areas was not recommended for direct use for drinking. However, the water quality assessed based on the WQI in Vinh Long province tends to be better than in water bodies in other provinces, such as Dong Thap [27] and Can Tho [26]. Surface water was observed in large rivers and land use for non-agricultural purposes better than in agricultural land [26,35]. Moreover, natural factors, such as flow rate, surface area, and low water exchange capacity of rivers in the inland area, are limited. In addition, wastewater treatment and pre-treatment systems in non-agricultural and residential land use areas and effective state management activities also protect surface water quality. Thereby, it can be seen that there is significant influence of land use on water bodies [5,28,36].

### 4.4. Ranking and Correlating Surface Water Quality Parameters

EC and coliform were the main contaminants in 14 parameters because of the highest entropy weights. Based on the entropy calculation, the parameters can be sorted in descending order as follows: EC > coliform > $N-NH_4^+$ > BOD and $N-NO_3^-$ > Fe > pH; TSS and DO > turbidity and COD > $P-PO_4^{3-}$ and *E. coli*. The physicochemical and biological parameters in this study were negatively impacted by human activities. In addition, the Entropy weight indicated the stability of the water quality for the considered parameters. However, this result was reported differently from that of Giao et al. [27], showing that *E. coli* was the most influential parameter in the water bodies of Dong Thap province. This may be due to the difference in terrain elevation and anthropogenic impacts. The close positive correlation of the parameters suggested that their origins were similar. This means that an increase in the concentration of one of the six parameters can increase the remaining contaminants (except Fe and EC). Typically, an increase in organic matter (BOD) leads to an increase in the concentrations of nutrients ($N-NH_4^+$ and $N-NO_3^-$). This correlation was also reported in the previous study [37]. On the other hand, this increase may lead to

increased respiration and decomposition of organic matter, reducing the DO concentration and negatively affecting water quality.

*4.5. Key Variables Influencing Surface Water Quality*

PC1 represents the significant influences of human activities on surface water, including both point- and nonpoint- sources of pollution. Point sources of pollution originate from pollutants from improper domestic waste disposal, while nonpoint sources are agricultural runoff containing fertilizers and pesticides. PC2 also showed the use of chemical fertilizers in agriculture and the leaching of $P-PO_4^{3-}$ and soil particles, increasing $P-PO_4^{3-}$ and TSS in surface water. PC3 represented the influence of wastewater from market areas scattered in the study area, where there was a high concentration of organic matters (food) and wastewater. Additionally, the loading level of pH and $Cl^-$ showed that the salinity factor also partly influenced the water quality in Vinh Long province. Therefore, the presence of $Cl^-$ can be of natural or anthropogenic origin. PC4 represents a natural process and surface runoff from acid sulfate areas, leading to the leaching of iron oxides. Erosion is also a phenomenon in water bodies that increases turbidity values in water [10]. This can be considered an activity that was contributing to the nonpoint sources of pollution in Vinh Long province. On the other hand, the positive load factor of temperature may be due to seasonal variations in weather or other industrial activities that can increase the surface water temperature. This will likely affect chemical reactions, reaction rates, aquatic life, and the suitability of the water for other uses [38].

According to the direction of these principal components, it can be concluded that trading activities in markets, industrial zones, domestic wastewater, pesticides, and fertilizers used in agricultural activities were the main determinant factors of water pollution caused by runoff. Therefore, these sources have a higher degree of influence than those from natural processes. The analysis of PCA, CA, and WQI also confirmed that the sources were linked to local activities. Some similar results were also reported [12,26].

**5. Conclusions**

The results showed that surface water quality in Vinh Long province in 2019 was impaired by organic pollution, nutrients, and microorganisms. $Cl^-$ was detected only at four locations, of which S41 showed signs of salinity. DA results revealed that only eight water quality parameters (pH, turbidity, TSS, EC, DO, $Cl^-$, *E. coli*, coliform) are required to distinguish water samples between three sampling periods. Spatial CA showed the complexity of water bodies and produced 22 groups from 63 sampling locations based on the similarity of water quality characteristics. The number of locations can be reduced from 63 to 51 locations. The results of WQI showed that the poorest water quality was noted in the in-field canals and was regionally divided due to the influence of land-use patterns. The present study also showed the dominance of microbial parameters, organic matter, and nutrients in the following order: EC > coliform > $N-NH_4^+$ > BOD and $N-NO_3^-$ > Fe > pH, TSS and DO > turbidity and COD > $P-PO_4^{3-}$ and *E.coli* and were mainly responsible for water pollution. The results of PCA suggested that the pollution sources could be due to the extensive use of chemicals in agriculture, domestic wastewater, and rural and urban market areas and partly saline intrusion.

**Author Contributions:** Conceptualization, N.T.G. and H.T.H.N.; methodology, N.T.G. and D.V.N.; software, H.T.H.N.; validation, N.T.G., H.T.H.N., T.H.D. and P.K.A.; formal analysis, H.T.H.N. and T.H.D.; investigation, P.K.A.; resources, N.T.G.; data curation, N.T.G.; writing—original draft preparation, H.T.H.N., D.V.N. and P.K.A.; writing—review and editing, N.T.G.; visualization, H.T.H.N.; supervision, N.T.G.; project administration, N.T.G. All authors have read and agreed to the published version of the manuscript.

**Funding:** This research received no external funding.

**Institutional Review Board Statement:** Not applicable.

**Informed Consent Statement:** Not applicable.

**Data Availability Statement:** Not applicable.

**Acknowledgments:** The authors would like to express our sincere attitude toward the Department of Natural Resources and Environment Vinh Long province for data provision. The scientific and personal views presented in this paper do not necessarily reflect the views of the data provider.

**Conflicts of Interest:** The authors declare no conflict of interest.

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
