# Peer review of "Spatiotemporal Variations in Physicochemical and Biological Properties of Surface Water Using Statistical Analyses in Vinh Long Province, Vietnam"

_water, doi:10.3390/w14142200_

Round 1
Reviewer 1 Report
I recommend the following corrections.
1)Lines 77-78
“Acid sulfate soils are dominant, even in deeper soil layers, so the acidity is low”.
Please verify this information, may be the pH of soil is low, but acidity is high in acid sulfate soils.
2)The formulas of calculus for WQII, WQIII and WQIIII were not given
3) Lines 220-224
The significance of the MPN abbreviation (most probable number) should be given.
4) Line 239
Table 2,
Please verify “Clorua” or introduce the signification of this term
In Figure 6, Figure 8
Amoni is probably Ammonium. Please replace with the right term or with the formula.

Author Response
Dear Reviewer,
The authors would like to thank for your valuable comments. We followed your comments and corrected the manuscript. Please find the details of the responses in the attachment.
Kind regards

Reviewer 2 Report
Dear authors,
your paper "Spatiotemporal Variations in Physicochemical and Biological Properties of Surface Water Using Statistical Analyses in Vinh Long province, Vietnam" is a well-structured thorough study, and which results are worthy of publishing in Water.
Only a few minor changes are required:
1. L14 and 63: Please switch to "organic matter"
2. L110: Please change the names of the parameters, or their chemical formulas to avoid confusion - N-NH4+ is ammonim nitrogen, NH4+ is ammonium ion, NH3 is amonia. Please choose the one relevant to your measurement. N-NO3- is nitrate nitrogen, NO3- is nitrate(s). Same comment. P-PO43- is phosphorus as orthophosphate, PO43- is (ortho)phosphate(s).
L112: Please explain what hand-held equipment is, or choose another term.. If you are referring to a portable device(s), please include the manufacturer, etc.
L112-113: Please provide additional information regarding the laboratory used - was it accredited according to ISO/IEC 17025? Please provide additional information for the Quality control of the data obtained - (C)RMs used, recoveries, etc.
Author Response

(The authors gave the same response as above.)
